# Inhibiting *Leishmania donovani* Sterol Methyltransferase to Identify Lead Compounds Using Molecular Modelling

**DOI:** 10.3390/ph16030330

**Published:** 2023-02-21

**Authors:** Patrick O. Sakyi, Samuel K. Kwofie, Julius K. Tuekpe, Theresa M. Gwira, Emmanuel Broni, Whelton A. Miller, Michael D. Wilson, Richard K. Amewu

**Affiliations:** 1Department of Chemistry, School of Physical and Mathematical Sciences, College of Basic and Applied Sciences, University of Ghana, Legon, Accra P.O. Box LG 56, Ghana; 2Department of Chemical Sciences, School of Sciences, University of Energy and Natural Resources, Sunyani P.O. Box 214, Ghana; 3Department of Biomedical Engineering, School of Engineering Sciences, College of Basic & Applied Sciences, University of Ghana, Legon, Accra P.O. Box LG 77, Ghana; 4Department of Biochemistry, Cell, and Molecular Biology, West African Centre for Cell Biology of Infectious Pathogens, College of Basic and Applied Sciences, University of Ghana, Accra P.O. Box LG 54, Ghana; 5Department of Parasitology, Noguchi Memorial Institute for Medical Research (NMIMR), College of Health Sciences (CHS), University of Ghana, Legon, Accra P.O. Box LG 581, Ghana; 6Department of Medicine, Loyola University Medical Center, Maywood, IL 60153, USA; 7Department of Molecular Pharmacology and Neuroscience, Loyola University Medical Center, Maywood, IL 60153, USA; 8Department of Chemical and Biomolecular Engineering, School of Engineering and Applied Science, University of Pennsylvania, Philadelphia, PA 19104, USA

**Keywords:** *Leishmania donovani*, sterol methyltransferase, pharmacophore, ergosterol biosynthesis, molecular docking, molecular dynamics simulation, in vitro studies

## Abstract

The recent outlook of leishmaniasis as a global public health concern coupled with the reportage of resistance and lack of efficacy of most antileishmanial drugs calls for a concerted effort to find new leads. The study combined *In silico* and in vitro approaches to identify novel potential synthetic small-molecule inhibitors targeting the *Leishmania donovani* sterol methyltransferase (*Ld*SMT). The *Ld*SMT enzyme in the ergosterol biosynthetic pathway is required for the parasite’s membrane fluidity, distribution of membrane proteins, and control of the cell cycle. The lack of *Ld*SMT homologue in the human host and its conserved nature among all *Leishmania* parasites makes it a viable target for future antileishmanial drugs. Initially, six known inhibitors of *Ld*SMT with IC_50_ < 10 μM were used to generate a pharmacophore model with a score of 0.9144 using LigandScout. The validated model was used to screen a synthetic library of 95,630 compounds obtained from InterBioScreen limited. Twenty compounds with pharmacophore fit scores above 50 were docked against the modelled three-dimensional structure of *Ld*SMT using AutoDock Vina. Consequently, nine compounds with binding energies ranging from −7.5 to −8.7 kcal/mol were identified as potential hit molecules. Three compounds comprising STOCK6S-06707, STOCK6S-84928, and STOCK6S-65920 with respective binding energies of −8.7, −8.2, and −8.0 kcal/mol, lower than 22,26-azasterol (−7.6 kcal/mol), a known *Ld*SMT inhibitor, were selected as plausible lead molecules. Molecular dynamics simulation studies and molecular mechanics Poisson–Boltzmann surface area calculations showed that the residues Asp25 and Trp208 were critical for ligand binding. The compounds were also predicted to have antileishmanial activity with reasonable pharmacological and toxicity profiles. When the antileishmanial activity of the three hits was evaluated in vitro against the promastigotes of *L. donovani*, mean half-maximal inhibitory concentrations (IC_50_) of 21.9 ± 1.5 μM (STOCK6S-06707), 23.5 ± 1.1 μM (STOCK6S-84928), and 118.3 ± 5.8 μM (STOCK6S-65920) were obtained. Furthermore, STOCK6S-84928 and STOCK6S-65920 inhibited the growth of *Trypanosoma brucei*, with IC_50_ of 14.3 ± 2.0 μM and 18.1 ± 1.4 μM, respectively. The identified compounds could be optimised to develop potent antileishmanial therapeutic agents.

## 1. Introduction

The challenges associated with control coupled with the frequent epidemic outbreak have rendered leishmaniasis an important public health problem [1,2,3]. Consequently, the presence of the ugly lesions after treatment and the severity of those caused by *Leishmania donovani* and *Leishmania infantum* in humans has rendered visceral leishmaniasis a global public health concern that needs urgent attention [1,3]. Currently, there are no vaccines and the number of chemotherapeutic options is limited [4,5]. Drugs including pentamidine, amphotericin B, paromomycin, miltefosine, and stibogluconate are ineffective and toxic, and the parasite has developed resistance, requiring renewed and concerted efforts to identify and develop new antileishmanial chemotypes [6,7]. Some efforts have been made towards finding new treatment options from natural products [8,9]; however, their structures and chemical synthesis are complex [10,11].

Synthetic small molecules with drug-like properties have been explored for the treatment of various ailments including leishmaniasis [12,13]. Recent studies show that about 60% of the drugs used for treating most diseases originated from synthetic small molecules [14,15]. The inherent properties of synthetic small molecules such as the ability to cross biological barriers and to modulate diverse biological targets render them viable drug candidates [16].

Target identification and validation are pivotal in drug design and the success of drug development depends primarily on choosing the right target. Studies in the design of antileishmanial compounds have identified some novel pathways and protein targets necessary for the parasite’s survival [17,18]. Some of these targets have been validated [19,20], while investigation on others is still ongoing. Ergosterol biosynthesis is involved in various biological functions including plasma membrane formation, membrane fluidity, distribution of membrane proteins, and control of the cell cycle [21]. Ergosterol is essential for optimal mitochondrion function in *Leishmania* parasites [22]. This pathway is catalysed by several enzymes with homologues in the human host and hence has the potential to cause off-target effects, mainly as a result of poor selectivity [23]. Sterol methyltransferase, which catalyses the transfer of the methyl group from S-adenosine-methionine to the C24 position of the sterol side chain during ergosterol biosynthesis, lacks a homologue in humans [22,24], and is thus considered an essential drug target.

Reports have shown that 22,26-azasterol suppresses the growth of *L. donovani* with a half-maximal inhibitory concentration (IC_50_) of 8.9 μM [25]. To improve the inhibitory effect of 22,26-azsterol, some analogues (Figure 1) were generated, and their activities were found to be dependent on the presence of ammonium or sulfonium functionality in the side chain [25]. Similarly, ezetimibe, imipramine, and simeprevir (Figure 1) have shown promising inhibitory activity against *L. donovani* by making morphological changes to the plasma and mitochondrion membranes with IC_50_ of 30, 28.6, and 51.49 μM, respectively [26,27,28]. A fragment-based de novo design was used to predict potential inhibitors against *L. donovani* sterol methyltransferase (*Ld*SMT) [29]. Though several hit compounds have been identified targeting this enzyme, only a few progressed to the clinical evaluation stage [30,31,32,33,34].

*In silico* techniques offer alternative platforms for the identification of novel hits in a timely and cost-effective manner [35,36]. These approaches have improved rational drug design and augmented the identification of potentially new drug candidates. This study, therefore, employs ligand-based pharmacophore virtual screening, molecular docking, molecular dynamics (MD) simulations, molecular mechanics Poisson–Boltzmann surface area (MM/PBSA), biological activity predictions, and in vitro studies to support the identification of potential inhibitors. The synthetic compound library was screened against *Ld*SMT followed by the elucidation of the mechanisms of binding of the proposed molecules.

## 2. Materials and Methods

The schematic workflow employed in the project is shown in Figure 2. First, six known inhibitors of *Ld*SMT with IC_50_ less than 10 μM were used to generate a pharmacophore model employing the merged feature embedded in LigandScout version 4.3 [37]. The pharmacophore model was then used to screen a library of 95,630 synthetic compounds from InterBioScreen limited. Ligands with pharmacophore fit scores above 50 were docked against a previously modelled structure of *Ld*SMT [29,38]. The selected hit compounds were evaluated via physicochemical, pharmacological, and toxicity profiles, as well as biological activity predictions, MD simulations, and MM/PBSA computations. Three potential lead compounds were then screened in vitro to validate the *In silico* studies.

### 2.1. Target Preparation

The three-dimensional (3D) structure of *Ld*SMT previously elucidated using Modeller version 10.2 [29,38] was used. Energy minimisation of the target protein was carried out by employing the Optimized Potential for Liquid Simulations (OPLS)/All Atom force field [39] via the Groningen Machine for Chemical Simulations GROMACS version 2018 (GROMACS 2018) [40,41]. Biovia Discovery Studio Visualizer v.19.1.0.18287 [42] was used to visualise the energy-minimised structure, remove water molecules, and solvate the protein, and then the result was saved in Protein Data Bank (pdb) format, which was later converted to AutoDock Vina’s [43] compatible pdbqt format.

### 2.2. Binding Site Prediction

The amino acid residues and the probable volume and area of the binding site of the modelled protein were determined using the Computed Atlas of Surface Topography of proteins (CASTp) [44] and Biovia Discovery Studio Visualizer v.19.1.0.18287 [42], as indicated in a previous study [29].

### 2.3. Ligand-Based Pharmacophore Virtual Screening

Six of the known inhibitors (Figure 1) of *Ld*SMT with IC_50_ less than 10 μM were employed in generating the pharmacophore model using LigandScout version 4.3 [37]. The structure data file (sdf) formats of the ligands were used via LigandScout’s Ligand-Based Modeling Perspective v.4.3 [37]. The default settings of OMEGA best were used to generate a maximum of 200 conformations per ligand.

### 2.4. Retrieval and Preparation of Chemical Library for Pharmacophore-Based Screening

A chemical library of 95,630 synthetic compounds was retrieved from InterBioScreen limited [45] and used for the pharmacophore-based virtual screening.

### 2.5. Pharmacophore-Based Virtual Screening of the Libraries

The 95,630 chemical entities retrieved from InterBioScreen limited were utilised for the pharmacophore screening using LigandScout v.4.3 [46], first by converting from “sdf” to “lbd” and then by screening against a validated pharmacophore model.

### 2.6. Validation of Pharmacophore Model and AutoDock Vina

Before performing the virtual screening, both the pharmacophore model and AutoDock Vina v.1.2.0 [43] were validated to assess their potential for distinguishing between active compounds and decoys.

#### 2.6.1. Pharmacophore Model Validation

The receiver operating characteristic (ROC) curve and enrichment factors (EFs) were used in validating the pharmacophore model in LigandScout v.4.3 [37]. The six inhibitors of SMT and their decoys were used to generate the ROC curve from where the area under the curve (AUC) and EFs were calculated.

#### 2.6.2. Validation of AutoDock Vina

The simplified molecular input line entry system (SMILES) of six of the known inhibitors with IC_50_ less than 10 μM served as active compounds for the generation of 50 decoys each using the Directory of Useful Decoys (DUD-E) [47]. The ROC curves generated via EasyROC v.1.3.3 [48] were used to validate the docking protocol of AutoDock Vina v.1.2.0 [43].

### 2.7. Molecular Docking Studies of Chemical Entities with Good Pharmacophore Fit Scores

AutoDock Vina [43] interfaced with PyRx v.0.8 [49], which employs empirical scoring functions, was used for the virtual screening. Ligands with good pharmacophore fit scores were obtained after screening the pharmacophore model against the library combined with 22,26-azasterol and the three known drugs (amphotericin B, miltefosine, and paromomycin). The energy-minimised protein (*Ld*SMT) was used for molecular docking via AutoDock Vina v.1.2.0 [43]. The charge, hydrogen bond network, and histidine protonation state of the protein were assigned after pdbqt conversion [32]. The grid box size (91.445 × 73.502 × 78.352) Å^3^ with the centre (72.200, 58.009, 13.302) Å was used with an exhaustiveness of 8 [29].

### 2.8. Characterisation of Mechanism of Binding

Biovia Discovery Studio Visualizer v.19.1.0.18287 [42] was employed in the elucidation of the protein–ligand interactions.

### 2.9. ADMET Properties and Drug-Likeness Assessment

The absorption, distribution, metabolism, excretion, and toxicity (ADMET) properties were determined using SwissADME [50] and the OSIRIS Property Explorer in DataWarrior [51]. This was carried out to evaluate the drug-likeness and the pharmacologic properties of the selected compounds.

### 2.10. Prediction of Biological Activity of Selected Compounds

The biological activity of the compounds was predicted using Prediction of Activity Spectra for Substance (PASS) [52]. The SMILES files of the molecules were used as inputs.

### 2.11. Molecular Dynamics Simulation Analyses of Protein and Protein–Ligand Complexes

A 100 ns MD simulation was adopted [29] for *Ld*SMT and *Ld*SMT–ligand complexes via a sample interval for configuration of 100 ps using GROMACS 2018 [40,41]. Initially, the topology and position restrain files for the protein were generated using OPLS/All Atom force field [39]. LigParGen [53] was employed to generate the topology and parameter files of the ligands. The unbound protein and protein–ligand complexes were centred in a cubic simulation box with a 1 nm distance from the edges to restrain particle movements that have the potential to cause effects on the surface atoms [54]. System solvation was carried out using a single-point charge and then neutralised to achieve a molarity of 0.15 M using NaCl. Each system was energy-minimised for 50,000 steps using the steepest descent method. A 100 ps equilibration was carried out using several substances, volume and temperature, and several substances, pressure, and temperature ensembles [54] to ensure a well-equilibrated system was achieved at 300 K and a pressure of 1 bar. The root-mean-square deviation (RMSD), root-mean-square fluctuation (RMSF), the radius of gyration (Rg), and surface accessible surface area (SASA) trajectories were plotted using Qtgrace [55]. MM/PBSA computations of the complexes were carried out using g_MM/PBSA, which calculates the free binding energy and the individual energy contributions of the residues [56].

### 2.12. Parasite Culture and In Vitro Effect of Compounds on L. donovani Promastigotes

*L. donovani* promastigotes (MHOM/SD/62/1S strain—Bei Resources NIAID, NIH) were used for the cell viability assay. The promastigotes were grown at 25 °C in M199 medium supplemented with 10% heat-inactivated foetal bovine serum, penicillin G sodium (100 g/mL), and streptomycin sulphate (100 g/mL), and subcultured every 72 h in the same medium at a density of 2 × 10^5^ cells/mL.

To determine the concentration of a compound that inhibits the growth of 50% (IC_50_) of the *Leishmania* parasite population, promastigotes at a density of 2 × 10^5^ cells were incubated in triplicate without or with the compounds at varying concentrations (100–0.781) μg/mL and kept at 25 °C for 72 h. The Alamar blue test was used to determine the vitality of the parasites as previously described [57]. At an excitation wavelength of 530 nm and an emission wavelength of 590 nm, fluorescence was measured using a Varioskan Lux Elisa microplate reader (Thermo Fisher Scientific, Waltham, MA, USA). The fluorescence counts were plotted against drug concentrations, and the IC_50_ was calculated using a dose–response curve via GraphPad Prism version 9 (GraphPad Software Inc., San Diego, CA, USA). The reference drug used in the assay is amphotericin B, a known drug for treating leishmaniasis.

### 2.13. In Vitro Culture and Cell Viability Assay for Trypanosoma brucei brucei GUTat 3.1

Similar to the approach described in Section 2.12, to determine the IC_50_ of the compounds against *Trypanosoma brucei brucei* GUTat 3.1 parasite population, the modifications were *trypanosomes* at a density of 4×10^3^ cells, which were kept at 37 °C for 72 h [58]. The control drug used for this assay was diminazene aceturate (Sigma-Aldrich, Kent, UK), a veterinary drug used to combat infections of *trypanosomes* in cattle.

## 3. Results and Discussion

The results from the study are presented in the following sections, including the pharmacophore-based design, molecular docking, ADMET profiling, prediction of biological activities of selected molecules, MD simulations, MM/PBSA computations, and biological activity evaluations.

### 3.1. Predictions of LdSMT Binding Site Residues

A binding site is a pocket or groove on a protein that serves as a site for a ligand or biological macromolecule to bind with specificity. Ligands bind at active sites to induce biochemical reactions including methylation, demethylation, and phosphorylation. CASTp [44], which employs Delaunay triangulation, alpha shape, and discrete flow methods to identify topographic features and measure areas and volumes, was used to predict the binding site of *Ld*SMT in previous work [29]. The predicted binding site volume was 446.632 Å^3^ and the area was 905.262 Å^2^, and the critical amino acid residues elucidated included Ala30, Asp108, Val109, Tyr206, Gly207, Trp208, Met210, Tyr220, Tyr275, Leu278, Glu324, Ser328, Leu329, Val330, and Val331 [29].

### 3.2. Pharmacophore Generation

For pharmacophore generation and the prediction of features, six *Ld*SMT inhibitors (22,26-azasterol and X1-X5) (Figure 1) with IC_50_ < 10 μM were used. In addition, all six ligands and their decoys were employed to validate the pharmacophore model using the merged feature of LigandScout version 4.3 [37], similar to previous studies [59]. A predicted score of 0.9144 was generated with features consisting of three hydrophobic interactions (H), one positive ionisable (PI), one hydrogen bond acceptor (HBA), and one hydrogen bond donor (HBD) (Figure 3).

### 3.3. Validation of Generated Pharmacophore Model

The ROC curve evaluates the ability of the model to efficiently distinguish between a compendium of active and inactive compounds [59]. The ROC curve was obtained by plotting a graph of 1—specificity (Equation (1)) on the *x*-axis against sensitivity (obtained using Equation (2)) on the *y*-axis [60]. Notably, 1—specificity is called false positive fraction (FPF) and the sensitivity is also known as the true positive fraction (TPF).
(1)$$\mathrm{FPF}=\frac{\mathrm{FP}}{\left(\mathrm{FP}+\mathrm{TN}\right)}\;$$
(2)$$\mathrm{TPF}=\frac{\mathrm{TP}}{\left(\mathrm{TP}+\mathrm{FN}\right)}$$
where FP denotes false positive, TN as true negative, TP as true positive, and FN as false negative.

Consequently, while a poor test has a ROC curve falling on the diagonal line, a perfect test has the ROC curve passing through the upper left corner with approximately 100% sensitivity and 100% specificity [60]. The ROC curve (Figure 4a) falls to the upper left corner, implying a reasonably valid pharmacophore model. The AUC ranges from 0 to 1, with values close to 1 signifying a perfect model with the ability to distinguish active from inactive compounds [59]. The six known inhibitors serving as active compounds were used to generate 300 decoys made up of 50 decoys from each. A total of 306 compounds comprising six active molecules and 300 decoys were screened via the 3D pharmacophore model and the AUC was computed as 1 (Figure 4a). The pharmacophore-guided approach was used to identify inhibitors of *Mycobacterium ulcerans* Cystathione γ-synthase MetB and an AUC of 0.7 was obtained after validation [59]. In addition, EF, which is the ratio of the number of identified active compounds within a pool of highly ranked hits to those randomly selected from the original dataset, was computed to evaluate the effectiveness of the pharmacophore model [61,62]. The EF was computed using Equation (3) [61].
EF = (N_hits sampled/_N_sampled_)/(N_total hits/_N_total_)(3)
where N_hits sampled_ is the number of identified true hits in the hit list, N_sampled_ is the number of compounds in the hit list, N_total hits_ is the number of hits in the dataset, and N_total_ is the total number of compounds in the dataset.

A pharmacophore model with EF less than 1 suggests the classification ability is low. Similarly, EF equal to one shows that the model has equal chances of classifying active and inactive compounds. In addition, an EF greater than one suggests the model has a greater potential of classifying active compounds. A pharmacophore model generated for the design of sigma-1 ligands was classified as poor due to the computed EF being close to 1 at 10% [63]. Similarly, a model with EF score above 1 classified the inhibitors significantly better [61]. Consequently, the EFs were obtained as 52.0, 26.0, 26.0, and 26.0 for 1%, 5%, 10%, and 100%, respectively, implying the effectiveness of the pharmacophore model in classifying hits amongst the dataset.

### 3.4. Validation of Docking Protocol

Most docking algorithms, including the Lamarckian genetic algorithm and the empirical scoring for free energy binding employed in AutoDock Vina, sometimes fail to accurately predict the pose and scoring functions, warranting validation [64]. The computed AUC under the ROC was used to evaluate the performance of AutoDock Vina v.1.2.0 [43] in distinguishing the active from inactive compounds. To validate the docking protocol, the six active compounds and 300 decoys were virtually screened against the *Ld*SMT to generate the ROC curve and the AUC was computed as 0.9997 (Figure 4b) [65,66]. The ROC curve (Figure 4b) aligns with the upper left corner of the graph, which signifies a reasonably excellent docking protocol. The AUC of 0.9997 suggests that AutoDock Vina v.1.2.0 [43] has a plausible ability to distinguish active compounds from decoys.

### 3.5. Pharmacophore-Based Virtual Screening of Library

Pharmacophore-based virtual screening is the generation of a 3D query by exploring the chemical features responsible for the biological activity from a reasonable number of structures for the identification of new chemotypes [67,68]. The surge in interest is due to the high enrichment of novel actives generated from a library of chemical entities [67]. The validated pharmacophore model was used as a 3D query to screen a chemical library of 95,630 synthetic compounds from InterBioScreen limited with a pharmacophore fit score of 50 as the threshold. STOCK6S-64914 and STOCK6S-47366 had the highest and least pharmacophore fit scores of 59.39 and 57.15, respectively, among the 20 shortlisted compounds (Table 1).

### 3.6. Molecular Docking Analysis

Molecular docking predicts the orientation and conformation of the ligand within the binding site of the target protein. Strategically, molecular docking is supposed to mimic a biological system, but being a computational-based approach [69], it often suffers certain drawbacks. In the current study, the effects and implications of solvent present in the protein that otherwise may affect the pose of the ligand [70] were not considered, as water molecules were deleted during protein preparation. MD simulation studies were carried out to ascertain the contribution of the solvent in the free binding energy of the protein–ligand complexes similar to previous studies [71,72]. In addition, the preparations of both the protein and the ligand before docking have the potential to introduce errors, leading to missing bonds and abnormal geometries. This occurs when converting from pdb or sdf to the AutoDock Vina format pdbqt. Visualisation analysis of the protein and ligands after preparation and file conversion before docking ensures that this does not affect the pose of the ligands. The exhaustiveness in the current study was set to default 8, which statistically may not represent the actual number of possible conformations of ligands in the binding pocket of the biological target. This is normally occasioned to save computational time and energy. Furthermore, the top-ranked pose was chosen as the best pose with the least binding energy. Interestingly, this is not always the case, as the top-ranked score may not necessarily represent the best pose. Despite these drawbacks, molecular docking has provided an ideal platform for studying molecular inclusions at the atomic level [73].

Molecular docking has been explored in the identification of lead compounds including gentisic acid 5-O glucoside [59], simeprevir [28], and bisindolylmaleimide IX [74] by targeting Buruli ulcer [59], visceral leishmaniasis [28], and SARS-CoV-2 [74], respectively. In all these instances, the binding energies associated with the protein–ligand complexes were used as part of the criteria to shortlist the hits. The 20 shortlisted compounds from the pharmacophore screening were docked into the catalytic domain [29] of the *Ld*SMT structure with binding energies comparable to or lower than that of 22,26-azasterol (−7.6 kcal/mol). The binding energies of nine compounds (Table 2) were also lower than those of the antileishmanial drugs amphotericin B (−5.3 kcal/mol), miltefosine (−5.0 kcal/mol), and paromomycin (−4 kcal/mol) (Table 2). The binding energies ranged from −7.5 to −8.7 kcal/mol with STOCK6S-07353, STOCK6S-16994, and STOCK6S-06707 having −7.5, −7.5, and −8.7 kcal/mol, respectively. The results showed that the nine selected compounds possess better binding energies and can be explored as further potential inhibitors.

### 3.7. Protein–Ligand Interaction

Molecular recognition due to protein–ligand interactions is the basis of most cellular mechanisms. The examination of this at the atomistic level provides insights into the design of small-molecule drugs. Molecular docking was undertaken to elucidate the mechanism of binding of the ligands as well as to identify small molecules with high cooperativity [75] and affinity to the *Ld*SMT. An evaluation of the 2D interactions [42] showed that the compounds including 22,26-azasterol and the three drugs interacted with amino acid residues Asp25, Ala28, Phe93, Phe100, Trp208, Asp218, Arg227, Ile228, Ala311, Glu320, and Leu322, consistent with previous studies [29,76]. Specific interactions in all of the complexes included *pi*–anion, *pi*–*pi* stacking, *pi*–alkyl, *pi*–sigma, carbon–hydrogen, and hydrogen bonds.

STOCK6S-06707, STOCK6S-55084, and STOCK6S-14893 did not form hydrogen bonds with any of the amino acid residues in the binding site. On the contrary, hydrogen bond interactions were observed for STOCK6S-65920 and STOCK6S-16994 via Ser330 and Thr338 (Table 2 and Appendix A). STOCK6S-06707 (−8.7 kcal/mol) only formed hydrophobic interactions with Asp25, Ala28, Aso31, Arg32, Phe307, and Ala311 (Table 2 and Figure 5). In addition, STOCK6S-64941 and STOCK6S-19430 formed two hydrogen bonds with active site residues (Table 2, Appendix A), while STOCK6S-84928 and STOCK6S-07353 formed three hydrogen bonds. Hydrophobic interaction analysis showed that Leu13, Trp208, Ile224, Tyr316, Lys313, and Leu322 (Table 2, Figure 5, and Appendix A) could be essential for the ligand binding of STOCK6S-84928, STOCK6S-07353, STOCK6S-64941, and STOCK6S-19430. Furthermore, STOCK6S-16994, STOCK6S-55084, and STOCK6S-65920 formed hydrophobic interactions with Phe93, Ile228, Glu192, and Arg195 (Table 2, Appendix A), which could enhance the protein–ligand stability. This result corroborates earlier studies demonstrating that amino acids between the ranges 15–100, 200–250, and 280–320 were critical for ligand binding [28,29,77].

The known inhibitor 22,26-azasterol (−7.6 kcal/mol) formed hydrogen bonds with Glu102 and Gly200 (Table 2 and Appendix A). Then, paromomycin, amphotericin B, and miltefosine formed five, four, and one hydrogen bond(s), respectively. The hydrophobic interactions formed by 22,26-azasterol and known drugs with Phe100, Lys198, Pro199, and Glu306 (Table 2 and Appendix A) may support the stability of the complexes.

### 3.8. ADMET Profiling

The drug-likeness of the nine selected compounds was evaluated using Lipinski’s rule of five (Ro5) [78,79]. According to the Ro5, a compound is classified as drug-like and orally active if it conforms to the following criteria: a molecular weight ≤500 Da, an octanol–water partition coefficient log *p* ≤ 5, HBD ≤ 5, and HBA ≤ 10 [79]. The Ro5 has been applied in many studies, especially during the shortlisting of hits [80,81,82]. The physicochemical properties and drug-likeness computed via SwissADME [50] showed that none of the compounds violated any of the Ro5 (Table 3). Orally active drugs, according to Veber’s rule, must possess a maximum of 10 rotatable bonds as well as a polar surface area of less than 140 Å^2^ [83]. All nine hits were predicted to have less than 10 and 140 Å^2^ for the number of rotatable bonds and polar surface area, respectively. STOCK6S-64941 had the highest number of rotatable bonds (6) and a polar surface area of 80.57 Å^2^. The results (Table 3) suggest that the selected compounds possess the potential to be orally active [83].

Molar refractivity (MR) measurement was conducted to assess the movement of the compounds through the body and their ability to reach their targets of inhibition at optimum concentrations [84]. MR for druggable candidates is recommended to be between 40 and 130 [29]. All of the selected compounds were found to possess MR within this range except STOCK6S-64941 with an MR of 130.16 (Table 3).

For drugs to be transported by the circulatory system to their site of activity, they are required to be soluble [85]. The computed solubility profile of the compounds showed them as soluble and hence possess the potential to reach their target sites at maximum concentrations (Table 3). Furthermore, the bioavailability score (BS), which measures the ability of drugs to reach systemic circulation, was also estimated [86]. The computed BS for the compounds was 0.55, similar to the reported studies (Table 3) [29,86].

The ability of a drug to cross the blood–brain barrier (BBB) to the brain is called BBB permeation [87]. Drugs with the ability to cross the BBB could bind to certain receptors to elicit the required pharmacological activities within the brain parenchyma. STOCK6S-55084, STOCK6S-64941, and STOCK6S-19430 (Table 3) were predicted not to be BBB permeants. Previous reports showed that *Leishmania* parasites infect and inflame the central nervous system (CNS) [88,89,90] and drugs with the potential of crossing the BBB could mediate in the elimination of the parasites in the CNS.

Next, we investigated the gastrointestinal absorption of the compounds. Together with 22,26-azatserol, they were predicted to possess high gastrointestinal absorption, suggesting a high probability of absorption into the bloodstream. On the contrary, the three drugs, amphotericin B, miltefosine, and paromomycin, were predicted to show low gastrointestinal absorption (Table 3). A previous study to identify potential inhibitors of *L. donovani* cell division cycle-2-related kinase 12 (CRK12) predicted compounds to possess high gastrointestinal absorption [32].

Furthermore, the toxicity profile (Table 4) of the compounds was evaluated using OSIRIS DataWarrior 5.0.0 [51]. The results showed that all of them were predicted not to be mutagenic or tumorigenic. On the other hand, STOCK6S-0670 was predicted as a high irritant, whilst STOCK6S-65920 and STOCK6S-55084 were found to possess low reproductive effects. Compounds STOCK6S-19430, STOCK6S-14893, and STOCK6S-16994 were predicted to possess high irritant and reproductive effects (Table 4).

### 3.9. Biological Activity Predictions of Selected Hit Compounds

The biological activities of the selected compounds were predicted [52] based on the structure–activity relationship. Compounds predicted with a probability of activity (Pa) greater than the probability of inactivity (Pi) have high chances of exhibiting the expected biological activities, hence could be considered for experimental validation [52,81].

All nine selected compounds were predicted to possess antineoplastic properties with Pa greater Pi, whilst STOCK6S-65920 and STOCK6S-55084 also act as CDK9 inhibitors (Appendix A). The anticancer drugs sunitinib, sorafenib, and lapatinib, which are known CDK inhibitors, were predicted as antileishmanial [30,31,32].

Furthermore, four of the selected hit compounds, STOCK6S-06707, STOCK6S-84928, STOCK6S-14893, and STOCK6S-16994, were predicted (Appendix A) to be dermatologic, suggesting that they could be developed as a potential treatment for post-kala-azar leishmaniasis [91,92]. The compounds STOCK6S-65920 and STOCK6S-55084 were also predicted to be inhibitors of lanosterol 14alpha demethylase. The drugs ketoconazole, fluconazole, itraconazole, and posaconazole are already in various clinical stages for leishmaniasis treatment targeting the *Leishmania* parasites lanosterol 14alpha demethylase [93,94].

STOCK6S-64941 and STOCK6S-07353 were predicted as potential aspulvinone dimethylallyl transferase and indole pyruvate C-methyltransferase inhibitors (Appendix A), respectively. The *Ld*SMT belongs to the family of methyltransferases to which aspulvinone dimethylallyl transferase and indole pyruvate C-methyltransferase also belong [95,96,97]. The predicted inhibition of these two targets suggests the potential of the selected hit compounds to suppress *L. donovani* sterol methyltransferase.

Furthermore, five of the selected hits comprising STOCK6S-06707, STOCK6S-55084, STOCK6S-19430, STOCK6S-14893, and STOCK6S-16994 were predicted for Alzheimer’s treatment (Appendix A) with Pa greater than Pi. A recent in vitro study identified perphenazine, a known drug for Alzheimer’s treatment, to possess antileishmanial potential with EC_50_ of 1.2 μg/mL [98]. Similarly, clomipramine, a drug for Alzheimer’s treatments, induced programmed cell death in *L. amazonensis* via a mitochondrial pathway disruption with an IC_50_ of 8.31 μM [99,100]. The predicted biological activities and mechanisms suggest that the selected compounds have potential antileishmanial scaffolds.

### 3.10. MD Simulations Analyses

Three of the hit compounds, STOCK6S-06707, STOCK6S-84928, and STOCK6S-65920, were selected for downstream MD simulation analysis. They possessed low binding energies, reasonable pharmacological profiles, and desirable biological activities. The control used for the MD was 22,26-azasterol. A 100 ns MD simulation was performed to assess the structural stability and the conformational dynamics of the *Ld*SMT–analogue complexes. The RMSDs were computed to ascertain the stability of the protein–ligand complexes (Appendix A). The complexes showed comparable stability to the unbound protein and protein–22,26-azasterol complex. The unbound *Ld*SMT and the protein–ligand complexes experienced a high volatility within the first 10 ns of the MD simulation with an RMSD fluctuation of 0.15 nm (Appendix A), which is classified as acceptable, since the drifting falls within the range 0.1–0.3 ns [101]. Equilibration was therefore reached afterward with a low RMSD fluctuation of ≤0.05nm. Remarkably, the RMSD results produced were corroborated by the RMSF values (Appendix A).

The probability distribution function (PDF) analysis revealed that all of the complexes, including the unbound protein, had only one peak, meaning the complexes and the unbound protein were stable with fewer deviations. This corroborates SARS-CoV-2 main protease (M^pro^) studies using malaria box compounds as hits [102] and the screening of inhibitors against *Leishmania donovani* 3-mercapto pyruvate sulfurtransferase [103]. The unbound protein, 22,26-azasterol, STOCK6S-84928, and STOCK6S-65920 complexes reached equilibrium with an average RMSD of 0.25 nm (Appendix A). In addition, the average RMSDs for the *Ld*SMT–STOCK6S-06707, *Ld*SMT–STOCK6S-84928, *Ld*SMT–STOCK6S-65920, *Ld*SMT–22,26-azasterol, and *Ld*SMT were 0.25 ± 0.03, 0.25 ± 0.03, 0.25 ± 0.02, 0.25 ± 0.03, and 0.25 ± 0.02, respectively, and were lower compared to an earlier study [29]. Overall, the compounds maintained a stable pose within the binding pocket of *Ld*SMT compared to the 22,26-azasterol.

The contributions of the amino acid residues to the stability of the protein–ligand complexes were analysed by computing RMSFs [104] (Appendix A). Despite the differences in the RMSFs, the fluctuation trends are similar among all of the complexes. The maximum fluctuations occurred for amino acid residues between 10–90, 100–160, 200–250, and 280–320 (Appendix A), and hence could be considered critical for ligand binding. Notably, from the PDF analysis (Appendix A) of the RMSF trajectory, the three hit compounds, *Ld*SMT–STOCK6S-06707, *Ld*SMT–STOCK6S-84928, *Ld*SMT–STOCK6S-65920, had average RMSFs of 0.13 ± 0.04, 0.15 ± 0.04, and 0.15 ± 0.04, respectively, compared to *Ld*SMT–22,26-azasterol (0.10 ± 0.04). Moreover, the unbound *Ld*SMT also had a similar average RMSF of 0.15 ± 0.04. Contrasting the RMSD and RMSF plots, the complexes were observed to be stable and comparable to the *Ld*SMT–22,26-azasterol complex.

The rigidness of the unbound protein and the complexes for the entire simulation period is determined by Rg [32]. A folded protein normally maintains a relatively steady Rg over a given simulation period when compared to the unfolded period. The computed Rg plot showed that the complexes maintained steady stability over the entire period with the *Ld*SMT–STOCK65-06707 and *Ld*SMT–STOCK6S-84928 showing the lowest average Rg of 1.975 ± 0.02 and 1.975 ± 0.01 nm, respectively (Appendix A). Similarly, 22,26-azasterol, STOCK6S-65920 complexes, and the unbound *Ld*SMT recorded respective averages of Rg of about 2.0 ± 0.01, 1.975 ± 0.03, and 2.0 ± 0.01 nm, respectively. As per the PDF analysis, all of the complexes and the native protein had only one peak. Both the unbound *Ld*SMT and *Ld*SMT–ligand complexes had an average Rg of 2 nm (Appendix A). Furthermore, the Rg graph (Figure 6a) indicates that the compactness of the *Ld*SMT–STOCK6S-06707, *Ld*SMT–STOCK6S-84928, and *Ld*SMT–STOCK6S-65920 was maintained and kept steady after complex formation.

To determine the effect of loose packing on solvent accessibility, the SASA of the entire protein was computed to provide information on the protein–solvent interactions [103,105]. Generally, high SASA values imply an increase in structural enlargement for the protein–ligand complexes under the influence of solvent surface charges, resulting in a more flexible and unstable conformation [105]. A complex of *L. donovani* 3-mercapto pyruvate sulfurtransferase and rutin complex had an average SASA value of 44 nm^2^, which was more compact than the unbound protein with an average SASA of 48 nm^2^ [103]. In the current study, no differences were observed (Appendix A) for the unbound protein, the three hits, and 22,26-azasterol complexes. The computed average SASA value for the unbound *Ld*SMT, 22,26-azasterol, and the identified lead complex were 160 nm^2^ (Appendix A). In addition, per the PDF SASA analysis, the unbound *Ld*SMT, as well as 22,26-azasterol, STOCK6S-06707, STOCK6S-84928, and STOCK6S-65920 complexes, had average SASA values of 155.0 ± 5.4, 160.0 ± 5.3, 158.0 ± 6.1, 160.0 ± 4.8, and 155 ± 6.4 nm^2^ (Appendix A), respectively. The hit compounds formed stable hydrophobic interactions with *Ld*SMT compared to that of 22,26-azasterol.

To better understand the binding interactions that influenced the binding energy of the various protein–ligand complexes, the MD simulation was carried out to calculate the number of HB formed for all the *Ld*SMT–analogue complexes. The number of HB formed were (6 ± 1), (5 ± 1), (3 ± 1), and (2 ± 1) (Figure 6) for the *Ld*SMT–STOCK6S-06707, *Ld*SMT–22,26-azasterol, *Ld*SMT–STOCK6S-84928, and *Ld*SMT–STOCK6S-65920 complexes, respectively. Altogether, the hydrogen and hydrophobic interactions showed strong binding of the hit compounds in the active site of the *Ld*SMT.

### 3.11. MM/PBSA Computation of Free Binding Energies

Comparatively, binding free energy calculations from simulation studies have been proven to be more accurate than their docking counterparts and can prioritise compounds more efficiently for experimental evaluation [106]. Therefore, this work further employed MM/PBSA in computing the binding free energies of the protein–ligand complexes. Notably, a high negative binding free energy of a complex signifies a strong affinity of the ligand to the target protein. Among the selected hit compounds, STOCK6S-06707 with the lowest binding energy of −8.7 kcal/mol from the docking studies also recorded the lowest binding free energy of −371.146 ± 2.105 kJ/mol (Table 5). While STOCK6S-84928 and STOCK6S-65920 showed binding free energies of −129.725 ± 4.799 and −149.899 ± 3.6 kJ/mol, respectively, their binding free energies were higher than 22,26-azasterol (−221.527 ± 3.716 kJ/mol). The lead compounds showed promising binding free energies comparable to 22,26-azasterol and therefore warrant experimental validation. The free binding energy was contributed by van der Waal (ΔG_vdW_), electrostatic (ΔG_ele_), polar solvation (ΔG_pol, sol_), and SASA (ΔG_SASA_). Among these energies, van der Waal, responsible for the embedded hydrophobic interactions between the residues in the binding pocket and lead compounds, was the major contributor to the binding free energy (Table 5), followed by electrostatic. The results presented in this work corroborate previous studies that have shown that both van der Waal and electrostatic energies were responsible for the stability of protein–ligand complexes [29,32]. Despite the role of entropy in computing the free energy of binding to protein–ligand complexes [107], the current study did not take into account this contribution, because entropy tends to produce negligible effects when comparing the binding strength of ligands inhibiting the same protein target [108] and could be computationally expensive.

Favourable interactions between the binding site residues of the target protein and ligand result in the stability of the complex, and this was assessed by the per-residue decomposition, which computes the energy contribution of each of the amino acids. Generally, amino acids that contribute energies greater than +5 kJ/mol or lower than −5 kJ/mol are regarded as critical for ligand binding [59]. From the docking studies, residues Asp25, Ala28, Phe93, Phe100, Trp208, Asp218, Arg227, Ile228, Ala311, Glu320, and Leu322 were found to be key for ligand binding. The MM/PBSA per-residue decomposition computations revealed that Asp25 and Trp208 (Figure 7) contributed energies of −20 and −48 kJ/mol, respectively, for the stability of the *Ld*SMT–STOCK6S-06707 complex. Furthermore, the amino acid Trp208 contributed an energy of −10 kJ/mol (Figure 6) for the stability of STOCK6S-65920. The 22,26-azasterol complex generated per-residue decomposition trajectory with Glu102, Phe194, Phe198, and Gly200 (Appendix A), contributing energies above and below the threshold critical for ligand binding. Similarly, the per-residue decomposition trajectory of the STOCK6S-84928 complex shows Trp208 to produce energy of −7 kJ/mol for ligand binding. An earlier study to identify the potential inhibitors of *Ld*SMT showed similar residues as being critical for ligand binding [29]. The amino acid residues within the binding pocket, including Asp25 and Trp208 of *Ld*SMT, contributed meaningful energies to the stability of the complexes.

### 3.12. In Vitro Evaluation of Identified Hits against Leishmania donovani

The antileishmanial properties of three purchasable hit compounds, STOCK6S-06707, STOCK6S-84928, and STOCK6S-65920, were tested in vitro. The effects of different concentrations (Appendix A) on the survival of *L. donovani* promastigotes were studied with amphotericin B treatment as a positive control (Table 6). After 72 h of treatments of *L. donovani* promastigote cultures with varying concentrations of the three compounds, effect-based dose finding analysis revealed that concentrations of 21.9 ± 1.5 μM, 23.5 ± 1.1 μM, and 118.3 ± 5.8 μM (Table 6) for STOCK6S-06707, STOCK6S-65920, and STOCK6S-84928, respectively, eliminated 50 % of *L. donovani* promastigotes. Amphotericin B exhibited IC_50_ of 6.56 ± 0.06 μM. Despite the significant inhibitory effects of the hit compounds, their potencies were lower compared to amphotericin B. This notwithstanding, the compounds can be optimised to improve potency.

Both *Leishmania* and *Trypanosoma spp*. belong to the family of kinetoplastids [109] and some compounds have been shown to inhibit the survival of both *Leishmania* parasites and *Trypanosoma spp.* [110,111,112]. Compound SQ10, which suppressed the growth of *L. donovani*, was also reported to possess activity against *T. cruzi* [112]. Moreover, 22,26-azasterol, which inhibits sterol methyltransferase, also inhibits *T. cruzi* and *T. brucei* [25]. The anti-trypanosomal effects of the three compounds against *T. brucei* with diminazene as a positive control (Table 6) were undertaken via in vitro testing. Two of the compounds showed anti-trypanosomal activity with IC_50_ values of 14.3 ± 2.0 μM (STOCK6S-84928) and 18.1 ± 1.4 μM (STOCK6S-65920). Despite the lower activity of the two compounds compared to diminazene, they still possess significant inhibitory effects against *T. cruzi* and are hence worthy of further studies.

## 4. Contribution to the Field

In addition to the limited number, the available chemotherapeutic agents for the treatment of visceral leishmaniasis are plagued with inefficiencies and chemoresistance [113]. The integrated *In silico* and in vitro study, therefore, augments the current efforts in discovering antileishmanial agents by identifying inhibitors with the potential to circumvent the activities of *Leishmania donovani*. The elucidation of the mechanisms of binding of the potential lead compounds provides the platform for the identification of other potent inhibitors for leishmaniasis treatment targeting sterol methyltransferase.

## 5. Conclusions

Sterol methyltransferase is expressed in all *Leishmania* parasites, but does not have human homologues, making it a potential target for drug design against leishmaniasis. This study used *In silico* techniques including pharmacophore-based virtual screening, molecular docking, and MD simulations, as well as in vitro approaches to identify potent inhibitors against *Ld*SMT. Three compounds, STOCK6S-06707, STOCK6S-84928, and STOCK6S-65920, had a high binding affinity with binding energies lower than those of known inhibitors including 22,26-azasterol (−7.6 kcal/mol) and the antileishmanial drugs amphotericin B (−5.3 kcal/mol), miltefosine (−5.0 kcal/mol), and paromomycin (−4.0 kcal/mol). Molecular dynamics simulations and mechanistic studies revealed the leads to interact with critical amino acids in the binding site in the attenuation of *Ld*SMT. The compounds were predicted as antileishmanial with acceptable pharmacological and toxicity profiles. STOCK6S-06707, STOCK6S-84928, and STOCK6S-65920 were shown to suppress promastigotes of *L. donovani* cultures with IC_50_ values of 21.9, 23.5, and 118.3 μM, respectively. When the hits were tested for their anti-trypanosomal activity against *T. brucei*, STOCK6S-84928 and STOCK6S-65920 were active with IC_50_ of 14.3 and 18.1 μM, respectively. The promising antileishmanial compounds identified could serve as a basis for the design of potent biotherapeutic moieties.

## Figures and Tables

**Figure 1 pharmaceuticals-16-00330-f001:**
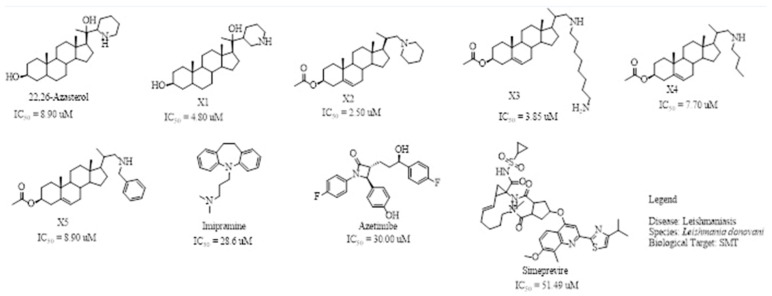
Chemical structures of some known inhibitors of *Leishmania donovani* sterol methyltransferase and their IC_50_ values. The biological target is sterol methyltransferase (SMT).

**Figure 2 pharmaceuticals-16-00330-f002:**
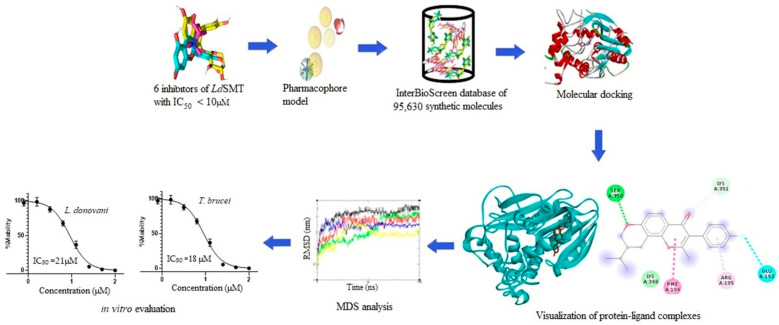
Methodology schema employed in the identification of antileishmanial compounds targeting *Ld*SMT. The workflow details the pharmacophore-based virtual screening, molecular docking, and MD simulation as well as in vitro validation of the selected potential antileishmanial and anti-trypanosomal chemotypes.

**Figure 3 pharmaceuticals-16-00330-f003:**
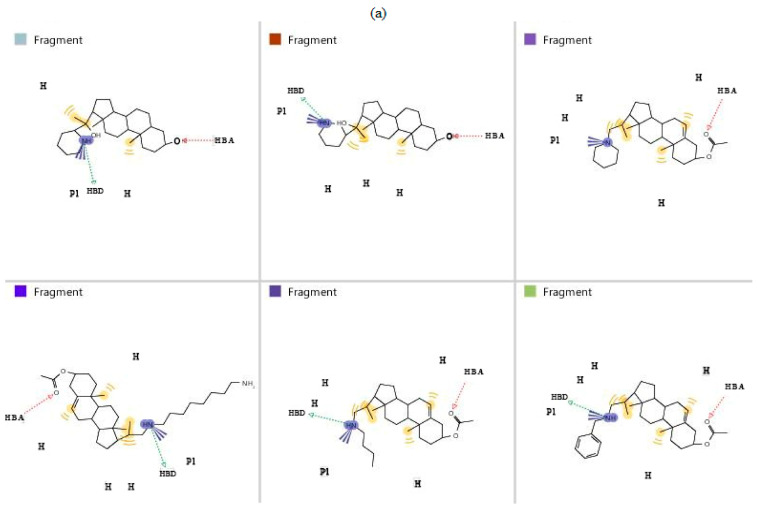
Pharmacophore features generated from (**a**) the six ligands comprising H, PI, HBA, and HBD; and (**b**) the pharmacophore model produced from merging the pharmacophore descriptors of all six inhibitors. Red balls indicate HBA, yellow balls denote H, green regions show HBD, and blue regions represent PI.

**Figure 4 pharmaceuticals-16-00330-f004:**
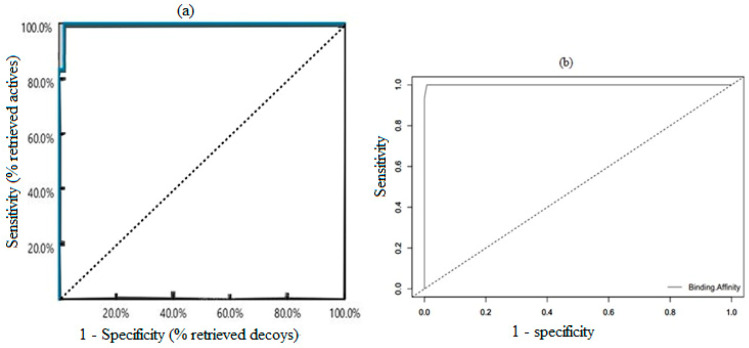
ROC curves for validation of the pharmacophore model and the docking protocols. (**a**) ROC curve of the selected pharmacophore model showing both the AUCs and EFs as measured in the top 1%, 5%, 10%, and 100% to be 52.0, 26.0, 26.0, and 26.0, respectively. (**b**) ROC curve generated after docking six active compounds and 300 decoys. The further away the median (that is, the dotted line) is from the curve, the better the model.

**Figure 5 pharmaceuticals-16-00330-f005:**
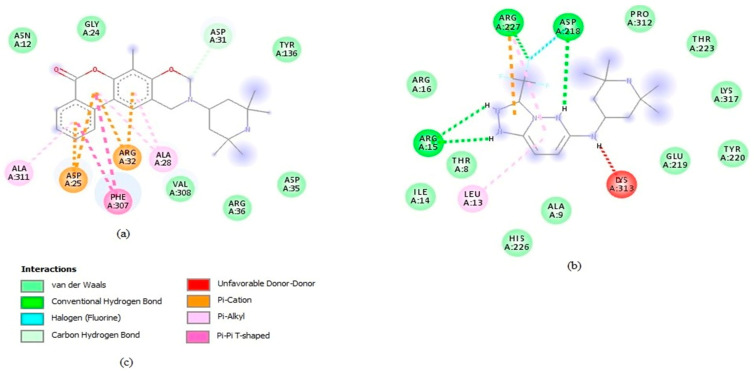
Two-dimensional interactions of the *Ld*SMT-hit complexes as visualised in Discovery Studio v.19.1.0.18287 [42]. (**a**) *Ld*SMT-STOCK6S-06707 complex, (**b**) *Ld*SMT-STOCK6S-84928 complex, and (**c**) legend of interactions.

**Figure 6 pharmaceuticals-16-00330-f006:**
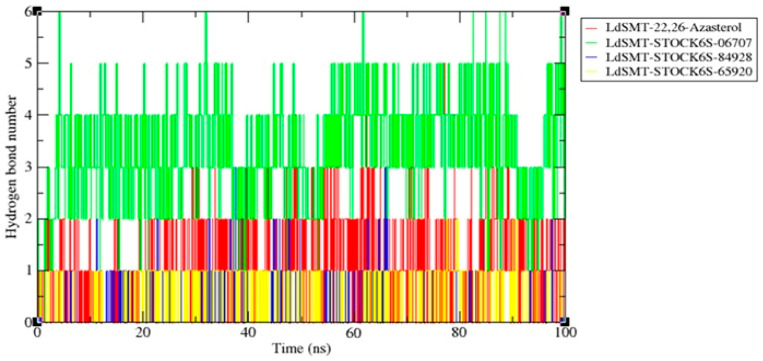
Number of hydrogen bond interactions of the protein–ligand complexes throughout the MD simulation period. Number of hydrogen bonds in *Ld*SMT–22,26-azasterol, *Ld*SMT–STOCK6S-06707, *Ld*SMT–STOCK6S-84928, and *Ld*SMT–STOCK6S-65920 complexes.

**Figure 7 pharmaceuticals-16-00330-f007:**
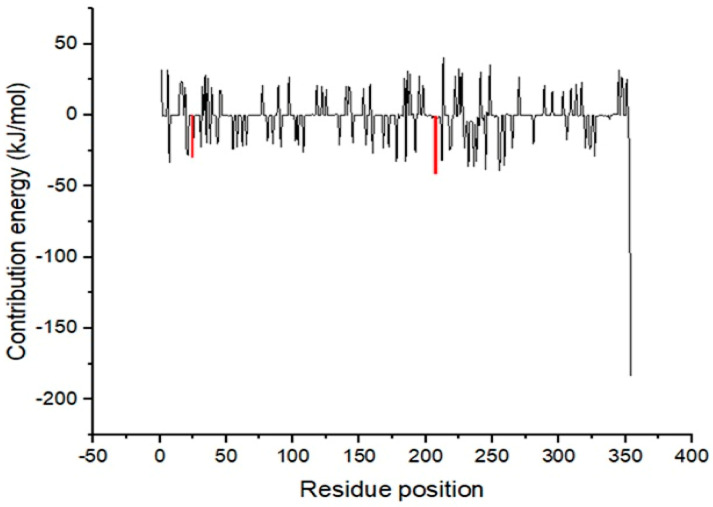
MM/PBSA computation of per-residue energy decomposition for the *Ld*SMT-STOCK6S-06707 complex.

**Table 1 pharmaceuticals-16-00330-t001:** Selected compounds with pharmacophore fit scores above 50 via pharmacophore-based virtual screening.

Compound ID	Pharmacophore Fit Score	Compound ID	Pharmacophore Fit Score
STOCK6S-64941	59.39	STOCK6S-14893	58.37
STOCK6S-43563	59.23	STOCK6S-06707	58.24
STOCK6S-07535	59.19	STOCK7S-11482	58.14
STOCK6S-33909	59.19	STOCK6S-16994	58.05
STOCK6S-39547	59.18	STOCK6S-64616	57.53
STOCK6S-19430	59.00	STOCK6S-65229	57.44
STOCK7S-75883	58.89	STOCK6S-63483	57.37
STOCK7S-14941	58.87	STOCK6S-55084	57.33
STOCK6S-84928	58.85	STOCK6S-65920	57.3
STOCK6S-47549	58.47	STOCK6S-47366	57.15

**Table 2 pharmaceuticals-16-00330-t002:** Binding energies of the selected compounds after molecular docking with generated intermolecular bonds.

Ligands	Binding Energy/kcal/mol	Binding Residues	
		Hydrogen Bonds	Hydrophobic Bonds
** 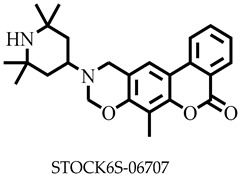 **	−8.7		Asp25, Ala28, Asp31, Arg32, Phe307, Ala311
** 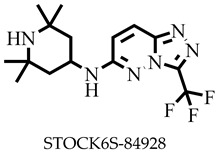 **	−8.2	Arg15, Asp28, Arg227	Leu13, Lys313
** 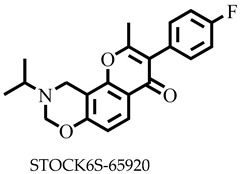 **	−8.0	Ser350	Glu192, Phe194, Arg195, Lys351
** 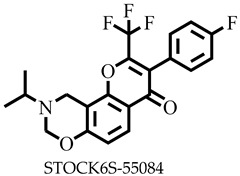 **	−7.9		Glu178, Trp208, His226, Ile228
** 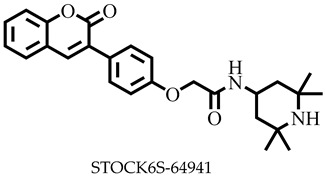 **	−7.8	Asp281, Ser284	Leu322
** 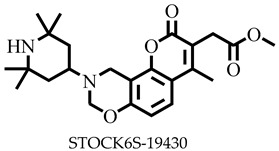 **	−7.7	Arg289, Arg295	Tyr316, Glu320
** 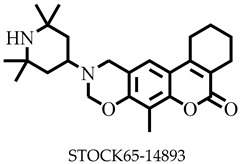 **	−7.6		Ala28, Phe307, Val308, Ala311
** 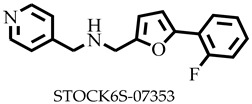 **	−7.5	Glu219, Glu229, Tyr275	Trp208, Ile224, Lys225, Ile228, Ile272
** 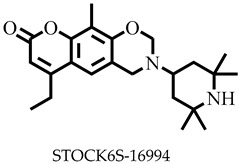 **	−7.5	Thr338	Phe93, Ile258
** 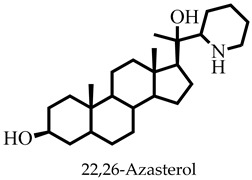 **	−7.6	Glu102, Gly200	Phe100, Lys198, Pro199
** 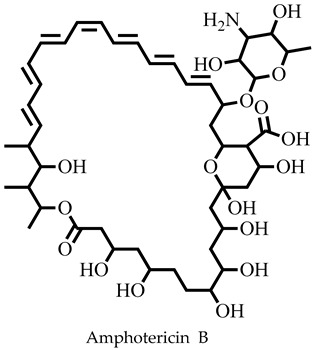 **	−5.3	Arg309, Asn299, Gly294, Leu288	
** 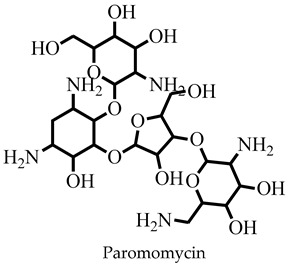 **	−5.0	Asp31, Arg32, Val308, Arg309, Leu310	Glu306
** 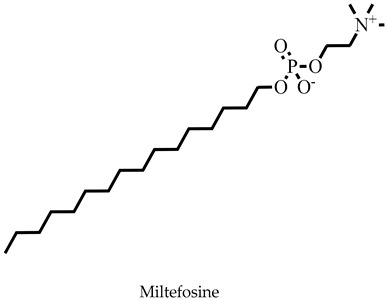 **	−4.0	Cys202	Phe100, Met101, Asp104, Asp172, Pro199, Gly200, Thr201, Tyr343, Ile344

**Table 3 pharmaceuticals-16-00330-t003:** Predicted physicochemical and pharmacological properties of the selected hit compounds. Molecular weight (MW), number of rotatable bonds (NRB), molar refractivity (MR), topological polar surface area (TPSA), gastrointestinal absorption (GI), blood–brain barrier (BBB), number of Ro5 violations (vRoF), bioavailability score (BS), and solubility score (SC) are presented.

Compound ID	MW (g/mol)	NRB	MR	TPSA (Å^2^)	LogS	SC	BS	GI	BBB	vRoF
STOCK6S-06707	406.52	1	129.05	54.71	−5.31	Moderate	0.55	High	Yes	0
STOCK6S-84928	342.36	3	88.22	67.14	−3.39	Soluble	0.55	High	Yes	0
STOCK6S-65920	353.39	2	103.42	42.68	−4.73	Moderate	0.55	High	Yes	0
STOCK6S-55084	407.36	3	103.46	42.68	−5.27	Moderate	0.55	High	No	0
STOCK6S-64941	434.53	6	130.16	80.57	−5.62	Moderate	0.55	High	No	0
STOCK6S-19430	428.52	4	127.41	81.01	−4.01	Moderate	0.55	High	No	0
STOCK6S-14893	410.55	1	128.98	54.71	−5.11	Moderate	0.55	High	Yes	0
STOCK6S-07353	282.31	5	78.96	38.06	−3.03	Soluble	0.55	High	Yes	0
STOCK6S-16994	384.51	2	121.32	54.71	−4.5	Soluble	0.55	High	Yes	0
22,26-Azasterol	403.64	2	125.08	52.49	−5.66	Moderate	0.55	High	Yes	1
Amphotericin B	924.08	3	239.06	319.61	−5.37	Moderate	0.55	Low	No	3
Miltefosine	407.57	20	115.9	68.40	−5.32	Moderate	0.55	Low	No	0
Paromomycin	615.63	9	133.56	347.32	−2.44	Soluble	0.55	Low	No	3

**Table 4 pharmaceuticals-16-00330-t004:** Toxicity profiles of the selected compounds predicted using OSIRIS DataWarrior 5.0.0.

Ligand	DataWarrior Predictions			
	Tumorigenic	Mutagenic	Irritant	Reproductive Effect
STOCK6S-06707	None	None	High	None
STOCK6S-84928	None	None	None	None
STOCK6S-65920	None	None	None	Low
STOCK6S-55084	None	None	None	Low
STOCK6S-64941	None	None	None	High
STOCK6S-19430	None	None	High	high
STOCK6S-14893	None	None	High	High
STOCK6S-07353	None	None	None	None
STOCK6S-16994	None	None	High	High
22,26-Azasterol	None	None	None	None
Amphotericin B	None	None	None	None
Miltefosine	None	None	None	None
Paromomycin	None	None	None	None

**Table 5 pharmaceuticals-16-00330-t005:** Contributions of van der Waal, electrostatic, polar solvation, and solvent-accessible surface area energies to the binding free energy of the protein–ligand complexes.

Complex	ΔG_vdW_ (kJ/mol)	ΔG_ele_ (kJ/mol)	ΔG_pol,sol_ (kJ/mol)	ΔG_SASA_ (kJ/mol)	ΔG_bind_ (kJ/mol)
STOCKIN6S-O6707	−232.978 ± 1.954	−983.173 ± 4.875	863.728 ± 3.447	−18.722 ± 2.602	−371.146 ± 2.105
STOCKIN6S−84928	−142.275 ± 5.679	−3.390 ± 0.417	29.163 ± 2.101	−13.223 ± 4.385	−129.725 ± 4.799
STOCKIN6S-65920	−217.243 ± 3.589	−6.439 ± 1.576	90.570 ± 4.465	−16.786 ± 1.646	−149.899 ± 3.600
22,26-Azasterol	−186.874 ± 2.143	−217.654± 0.546	199.212± 4.701	−16.211 ± 1.632	−221.527 ± 3.716

**Table 6 pharmaceuticals-16-00330-t006:** Results from the in vitro studies of the selected hit compounds against the promastigotes of *Leishmania donovani* and *Trypanosoma brucei*.

Compound	*Leishmania donovani* IC_50_ (μM) ± SD	*Trypanosoma brucei* (IC_50_) (μM) ± SD
STOCK6S-65920	23.5 ± 1.1	18.1 ± 1.4
STOCK6S-06707	21.9 ± 1.5	NA
STOCK6S-84928	118.3 ± 5.8	14.3 ± 2.0
Amphotericin B	6.56 ± 0.06	-
Diminazene	-	0.1 ± 0.02

NA = not available.

## Data Availability

Not applicable.

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
