# Peer review of "Inhibiting Leishmania donovani Sterol Methyltransferase to Identify Lead Compounds Using Molecular Modelling"

_pharmaceuticals, 2023, doi:10.3390/ph16030330_

Round 1

Reviewer 1 Report

The paper is well presented and has sufficient data as well as description of the methodology as to render possible the understanding of the data.

Minor revisions in style may be useful, however I would live this to the editor to judge.

My proposal for improvement of the paper is regarding the quality of the figures:

Figures 2, 3, 5 to 8 and 11 are of very poor quality, compromising the quality of the work. I suggest fusing figures 3 and 4 into a single figure and considering if they are really necessary for the understanding of the results.

Author Response

Reviewer #1

  1. Improvement in the quality of the figure

Response: Figure quality has been improved.

  1. Merging figure 3 and 4

Response: Figure 3 and 4 merged and is now Figure 3a and 3b

Reviewer 2 Report

the manuscript is of scientific interest and describes relevant information on possible antileishmanicidal candidates

Author Response

Reviewer #2

the manuscript is of scientific interest and describes relevant information on possible antileishmanicidal candidates

No corrections suggested by this reviewer

Reviewer 3 Report

The article combines in silico and in vitro approaches to identify new potential synthetic small molecule inhibitors targeting Leishmania donovani sterol methyltransferase (LdSMT). The study's relevance is justified by the fact that the recent view of leishmaniasis as a global public health problem, combined with reports of resistance and lack of efficacy of most anti-leishmania drugs, requires a concerted effort to find new clues. The LdSMT enzyme in the ergosterol biosynthetic pathway is essential for parasite membrane fluidity, distribution of membrane proteins, and cell cycle control. The lack of a LdSMT homolog in the human host and its conserved nature among all Leishmania parasites makes it a viable target for future anti-leishmania drugs. Therefore, six known LdSMT inhibitors with IC50 < 10 µM were initially used to create a pharmacophore model with a score of 0.9144 using LigandScout. This model was validated and then used to screen a synthetic library of 95630 compounds obtained from Interbioscreen limited. Twenty compounds with a pharmacophore fit score greater than 50 were docked to a simulated 3D LdSMT structure using AutoDock Vina. Therefore, nine compounds with binding energies ranging from -7.5 to -8.7 kcal/mol were identified as potential hit molecules, three of which, STOCK6S-06707, STOCK6S-84928, and STOCK6S-65920 with corresponding binding energies of -8 .7, -8.2 and -8.0 kcal/mol, lower than 22,26-azasterol (-7.6 kcal/mol), a known inhibitor of LdSMT, were chosen as likely lead molecules. Binding mechanism studies and molecular dynamics modeling showed that the three compounds interact with critical residues in the target protein's binding pocket.

Despite the satisfactory quality of the article, some shortcomings need to be corrected.

  1. The aim of the paper should be defined.
  2. Swapping the places of the Results and Materials and Methods sections are recommended.
  3. Formulas are parts of sentences. The correct punctuation and formatting should be used.
  4. The ROC curves presented in Figures 3 and 4 should be described in detail in the text.
  5. The quality of Figure 5 should be increased. It is impossible to read the legend.
  6. The author's contribution to the field should be highlighted.

In summarizing my comments, I recommend that the manuscript is accepted after minor revision. 

Author Response

Reviewer #3

  1. The aim of the paper should be defined

Response: The aim of the work is to employ ligand-based pharmacophore virtual screening in the identification of potential inhibitors of Leishmania donovani sterol methyltransferase (lines 105 - 109).

  1. Swapping the places of the Results and Materials and Methods sections are recommended.

Response: Results and Materials and Methods swapped.

  1. Formulas are parts of sentences. The correct punctuation and formatting should be used.

Response: Correct punctuation and formatting of equations is effected throughout the manuscript.

  1. The ROC curves presented in Figures 3 and 4 should be described in detail in the text.

Response: Details of ROC curves is provided in section 2.3, line 167 – 175 and line 214 – 215.

  1. The quality of Figure 5 should be increased. It is impossible to read the legend.

Response: The quality of Figure 5 now Figure 4 has been improved.

  1. The author's contribution to the field should be highlighted.

Response: Author’s contribution is highlighted in section 4 line 702 – 709.

Reviewer 4 Report

Comments on pharmaceuticals-2009824:

The current paper investigates the protein-ligand binding case involving Leishmania donovani Sterol Methyltransferase, with the title including the word ‘molecular modelling’. However, I find the quality of modelling rather weak, and significant improvements are required before consideration of publication. The presentation is also quite poor, degrading the readability of the manuscript. Below, I provide some detailed comments.

A lot of abbreviations are defined but not mentioned later. For instance, SAR is defined in the first paragraph of section 2.7 but is never mentioned later. In that case, they should not be defined at all. On the other hand, many abbreviations are defined multiple times. An example is MD, which is defined in the first paragraph of section 2 and also in the first paragraph of section 2.8. The authors should really carefully check this issue.

The RMSD reported in Fig. 6a is too huge. This normally relates to the inappropriate setup of the simulated system. If the system cannot be maintained in equilibrium fluctuations, systematic drifts would be observed. This fact can to some extent be validated in the systematic drift of Rg in Fig. 7a and Fig. 8a. The authors should either re-setup their system and re-run all the simulations to achieve equilibrium sampling, or lengthen the simulation time to see when these observables can really reach a plateau. Without these checks, the solidity of the current simulation outcome cannot be verified, and I cannot recommend publications in any journal.  

Further, the reported RMSF in Fig. 6c is meaningless and totally not informative. I strongly recommend the authors to move the whole Fig. 6, Fig. 7 and Fig. 8 to supporting material to shorten the main text and improve the readability of the paper. The current 32-page manuscript is too long compared with scientifically important contents reported in this work.

Instead of only reporting IC50, it would be more beneficial to explicitly provide the binding affinities.

Fig. 10 is also useless. The IC50 values are already reported in other Figures and Tables. These titration curves should also be moved to the supporting material.

Merging Figures. The distributions of observables in different systems are compared in Fig. 6b, Fig. 7b and Fig. 8b, but different systems are plotted separately, which significantly hinders the comparison between different systems. The authors should plot the distributions of different systems in the same plot in order to really enable a direct comparison.  

Many details of the computational study are absent. For example, which scoring function is used in molecular docking? What is the sampling interval for configurations during molecular simulation?

Author Response

Reviewer #4

  1. A lot of abbreviations are defined but not mentioned later

Response: Corrections effected throughout the manuscript. For example, Molecular Dynamics on line 544 was replaced by MD and (SAR) on line 511 was removed.

  1. The RMSD reported in Fig. 6a is too huge

Response: A re-run of the simulations resulted in the equilibration of the complexes reflective of the plateau of the RMSD plot leading to a revised average RMSD values.

  1. Move the whole Fig. 6, Fig. 7 and Fig. 8 to supporting material to shorten the main text and improve the readability of the paper.

Response: Figures 6, 7 and 8 were moved to supplementary data and are now supplementary Figures 2, 4 and 5 respectively.

  1. Instead of only reporting IC50, it would be more beneficial to explicitly provide the binding affinities.

Response: Binding free energies of the complexes and per-residue decompositions are provided in section 3.9:

  1. These titration curves should also be moved to the supporting material.

Response: Titration curves have been moved to supplementary data (Supplementary Figure 4)

  1. The authors should plot the distributions of different systems in the same plot in order to really enable a direct comparison.

Response: The distribution graphs are merged and moved to supplementary (Figures 2, 3, 4 and 5a) except for SASA which becomes messy after they were superimposed.

  1. Which scoring function is used in molecular docking? What is the sampling interval for configurations during molecular simulation?

Response: Autodock Vina scoring function is provided in section 3.7 line 769 and sampling interval for configuration was added (Section 2.11 line 203)

Reviewer 5 Report

The manuscript titled "Inhibiting Leishmania donovani Sterol Methyltransferase to Identify Lead Compounds Using Molecular Modelling" was an interesting read and very well-structured. It lacks certain points which can added/modified:

1) Several typos were observed and needs to corrected.

For eg. 'prefect' instead of 'perfect' on Line168

2) Authors have mentioned 'dynamic' instead of 'dynamics' in most parts of the manuscript.

3) Additional MM/PBSA calculations should be carried out in order to evaluate energy decomposition plot and binding free energy.

4) The figures in the main manuscript are not clear and should be improved.

Author Response

Reviewer #5

  1. Several typos observed which needs to be corrected

Response: Corrections effected in manuscript.

  1. Authors mention ‘dynamic’ instead of dynamics

Response: Dynamic has been changed to dynamics throughout the manuscript.

  1. Additional MM/PBSA calculations should be carried out

Response: Additional MMPBSA calculations were carried out and the results are provided section 2.9 and Table 5.

  1. Figures in the main manuscript needs improvement

Response: The Figures in the manuscript have been improved

Round 2

Reviewer 4 Report

Comments on pharmaceuticals-2009824.R1:

The current manuscript seems to be a minorly revised version that is still below the standard due to its large amount of presentation and scientific issues. In answering reviewers’ comments, all details of comments should be properly responded in a point-to-point manner and the manuscript should be revised accordingly. However, the current review response cuts my whole comment into segmented pieces with ignorance, and a lot of recommended points are not properly considered. Below are the missing points in the review response.   

First, my whole 78-words comment on the abbreviation issue is ‘abbreviated’ to 11 words by the authors in response, and the examples of problematic abbreviations are simply ignored. For example, although I have commented that the abbreviation SAR is defined but not never cited and in this case it should not be defined at all, the authors simply ignore this critical comment and also throw it away in their response. For such a careless revision, I believe that none of international journals with reputations would consider it as a properly responded response or carefully revised revision.

Second, my comments on the systematic drifting of Rg and RMSD are rather precise, but the authors carelessly read my comments and only take the large-RMSD point. As I emphasized in the previous review, the systematic drifting of structural observables suggests that the system is not under equilibrium and thus the sampling results are questionable. However, in the revised version, the RMSD is still rather huge, and the equilibrium-sampling behavior is still not secured.

The authors should know that the reviewers are volunteering their time to provide constructive comments to improve the quality of their manuscript, and the authors should take the reviewers’ comments in the spirit in which they are intended and carefully modify their paper.

Aside from many comments that are not properly responded in the previous round of review, there are many new problems identified in this new round of review.

The first problem is the huge uncertainty of MM/GBSA estimates. The binding free energies reported in Table 5 have statistical uncertainties > 50 kJ/mol, which are much larger than thermal fluctuations kBT ~2.5 kJ/mol and are unacceptably large. The huge fluctuation of ~50 kJ/mol is in magnitude similar to significant conformational changes such as the unbinding of protein-ligand complex. Numerical data of such low quality should not be published in any journal. Improvements in this aspect are necessary. Normally, we expect the numerical uncertainty to be decreased to a certain level in free energy calculations such as 1 kcal/mol (see references e.g., Journal of Chemical Information and Modeling 61 (1), 284-297).

There are also some other problems about the MMPBSA calculations in terms of presentation and calculation details. The MMPBSA method as a popular end-point free energy calculation method is often written as MM/PBSA. The free energy of binding is estimated with gas-phase interaction energies, the solvation part and also the entropic component. It seems unclear to me and also experienced researchers in this field why the entropic contribution is simply ignored without justification. There are many scientific reports emphasizing the importance of this contribution, see e.g., The Journal of chemical physics 146 (12), 124124. Performing entropy-involved calculations in end-point methods is now rather routine and should be properly considered.  

The uncertainty of RMSF is not properly defined. All statistical observables computed with molecular simulation should have some estimates of statistical uncertainties, including RMSF.

The binding pose analyzed for protein-ligand interaction is obtained from molecular docking. I would clarify that the aims of docking are mainly screening those non-binders with very low affinities and obtaining a preliminary guess of the bound structure of protein-ligand complexes. The low accuracy and non-exhaustiveness of conformational sampling in molecular docking often make noticeable deviations/differences between the docked pose and the real low-free-energy minimum. In modern MD simulations, the gold standard or the most reliable technique is to perform a thorough conformational search with accurate all-atom force field and enhanced sampling techniques. Some recent examples can be found in Carbohydrate Polymers 297, 120050. Without such a scan, it is difficult to say whether the binding pose is reliable or not. Therefore, to consolidate the binding-pose investigation in the current work, the authors should include some paragraphs discussing the limitation and possible pitfalls concerning the binding pose obtained from molecular docking.

Although I would like to consider a majorly revised version for publication, the authors are expected to really do some solid works at this stage to improve the scientific and presentation quality of their manuscript.

Author Response

We deeply regret the impression created by our first revision. The authors accept the comments of the reviewer in good faith and have responded  to them accordingly. Please see the response attached. For this current revision, authors have responded in a point-to-point manner. The manuscript has also been thoroughly revised as per the reviewer’s comments.

Round 3

Reviewer 4 Report

Comments on pharmaceuticals-2009824.R2:

The scientific rigor and presentation quality have been improved significantly in the second round of revision, and the current paper seems close-to-average in modern computational investigations of protein-ligand binding. As it has experienced two rounds of reviews, I would suggest acceptance this time. However, it should be noted that there are still many improvable points in many aspects, and the authors should endeavor to prepare their future papers of similar quality at the submission stage.

Author Response

We are grateful for the acceptance and suggestions. The content of the manuscript has been significantly improved. The English language, grammar and sentence structures have been improved considerably.
